# A neural mechanism for contextualizing fragmented inputs during naturalistic vision

Daniel Kaiser[1,2]*, Jacopo Turini[2,3], Radoslaw M Cichy[2,4,5]

[1]Department of Psychology, University of York, York, United Kingdom; [2]Department of Education and Psychology, Freie Universität Berlin, Berlin, Germany; [3]Institute of Psychology, Goethe-Universität Frankfurt, Frankfurt am Main, Germany; [4]Berlin School of Mind and Brain, Humboldt-Universität Berlin, Berlin, Germany; [5]Bernstein Center for Computational Neuroscience Berlin, Berlin, Germany

**Abstract** With every glimpse of our eyes, we sample only a small and incomplete fragment of the visual world, which needs to be contextualized and integrated into a coherent scene representation. Here we show that the visual system achieves this contextualization by exploiting spatial schemata, that is our knowledge about the composition of natural scenes. We measured fMRI and EEG responses to incomplete scene fragments and used representational similarity analysis to reconstruct their cortical representations in space and time. We observed a sorting of representations according to the fragments' place within the scene schema, which occurred during perceptual analysis in the occipital place area and within the first 200 ms of vision. This schema-based coding operates flexibly across visual features (as measured by a deep neural network model) and different types of environments (indoor and outdoor scenes). This flexibility highlights the mechanism's ability to efficiently organize incoming information under dynamic real-world conditions.

DOI: https://doi.org/10.7554/eLife.48182.001

*For correspondence:
danielkaiser.net@gmail.com

**Competing interests:** The authors declare that no competing interests exist.

## Introduction

During natural vision, the brain continuously receives incomplete fragments of information that need to be integrated into meaningful scene representations. Here, we propose that this integration is achieved through contextualization: the brain uses prior knowledge about where information typically appears in a scene to meaningfully sort incoming information.

A format in which such prior knowledge about the world is represented in the brain is provided by schemata. First introduced to philosophy to explain how prior knowledge enables perception of the world (*Kant, 1781*), schemata were later adapted by psychology (*Barlett, 1932*; *Piaget, 1926*) and computer science (*Minsky, 1975*; *Rumelhart, 1980*) as a means to formalize mechanisms enabling natural and artificial intelligence, respectively.

In the narrower context of natural vision, scene schemata represent knowledge about the typical composition of real-world environments (*Mandler, 1984*). Scene schemata for example entail knowledge about the distribution of objects across scenes, where objects appear in particular locations across the scene and in particular locations with respect to other objects (*Kaiser et al., 2019a*; *Torralba et al., 2006*; *Võ et al., 2019*; *Wolfe et al., 2011*).

The beneficial role of such scene schemata was first investigated in empirical studies of human memory, where memory performance is boosted when scenes are configured in accordance with the schema (*Brewer and Treyens, 1981*; *Mandler and Johnson, 1976*; *Mandler and Parker, 1976*).

Recently however, it has become clear that scene schemata not only organize memory contents, but also the contents of perception. For example, knowledge about the structure of the world can be used to generate predictions about a scene's content (*Bar, 2009*; *Henderson, 2017*), or to efficiently organize the concurrent representation of multiple scene elements (*Kaiser et al., 2014*; *Kaiser et al., 2019b*). This position is reinforced by behavioral studies demonstrating a beneficial role of schema-congruent naturalistic stimuli across a variety of perceptual tasks, such as visual detection (*Biederman et al., 1982*; *Davenport and Potter, 2004*; *Stein et al., 2015*) and visual search (*Kaiser et al., 2014*; *Torralba et al., 2006*; *Võ et al., 2019*).

Here, we put forward a novel function of scene schemata in visual processing: they support the contextualization of fragmented sensory inputs. If sensory inputs are indeed processed in relation to the schema context, scene fragments stemming from similar typical positions within the scene should be processed similarly and fragments stemming from different positions should be processed differently. Therefore, the neural representations of scene fragments should be sorted according to their typical place within the scene.

We tested two hypotheses about this sorting process. First, we hypothesized that this sorting occurs during perceptual scene analysis, which can be spatiotemporally pinpointed to scene-selective cortex (*Baldassano et al., 2016*; *Epstein, 2014*) and the first 250 ms of processing (*Cichy et al., 2017*; *Harel et al., 2016*). Second, given that schema-related effects in behavioral studies (*Mandler and Parker, 1976*) are more robustly observed along the vertical dimension, where the scene structure is more rigid (i.e., the sky is almost always above the ground), we hypothesized that the cortical sorting of information should primarily occur along the vertical dimension.

To test these hypotheses, we used a novel visual paradigm in which participants were exposed to fragmented visual inputs, and recorded fMRI and EEG data to resolve brain activity in space and time.

## Results

In our study, we experimentally mimicked the fragmented nature of naturalistic visual inputs by dissecting scene images into position-specific fragments. Six natural scene images (*Figure 1a*) were each split into six equally-sized fragments (three vertical × 2 horizontal), resulting in 36 conditions (six scenes × 6 fragments). In separate fMRI (n = 30) and EEG (n = 20) experiments, participants viewed these fragments at central fixation while performing an indoor/outdoor categorization task to ensure engagement with the stimulus (*Figure 1b*). Critically, this design allowed us to investigate whether the brain sorts the fragments with respect to their place in the schema in the absence of explicit location differences (*Figure 1c*).

To quantify the sorting of fragments during cortical processing we used spatiotemporally resolved representational similarity analysis (*Cichy et al., 2014*; *Kriegeskorte et al., 2008*). We first extracted representational dissimilarity matrices (RDMs) from the fMRI and EEG data, which indexed pairwise dissimilarities of the fragments' neural representations (for details on RDM construction see *Figure 2—figure supplement 1*). In the fMRI (*Figure 2a*), we extracted spatially-resolved neural RDMs from scene-selective occipital place area (OPA) and parahippocampal place area (PPA), and from early visual cortex (V1) (for temporal response profiles in these regions see *Figure 2—figure supplement 2*). In the EEG (*Figure 2b*), we extracted time-resolved neural RDMs from −200 ms to 800 ms relative to stimulus onset from posterior EEG electrodes (for other electrode groups see *Figure 2—figure supplements 3–5*).

We then quantified schema effects using separate model RDMs for horizontal and vertical locations (*Figure 2c*). These location RDMs reflected whether pairs of fragments shared the same location or not. We additionally constructed a category model RDM, which reflected whether pairs of fragments stemmed from the same scene or not.

Critically, if cortical information is indeed sorted with respect to scene schemata, we should observe a neural clustering of fragments that stem from the same within-scene location – in this case, the location RDM should predict a significant proportion of the representational organization in visual cortex.

To test this, we modeled neural RDMs as a function of the model RDMs using general linear models, separately for the fMRI and EEG data. The resulting beta weights indicated to which degree

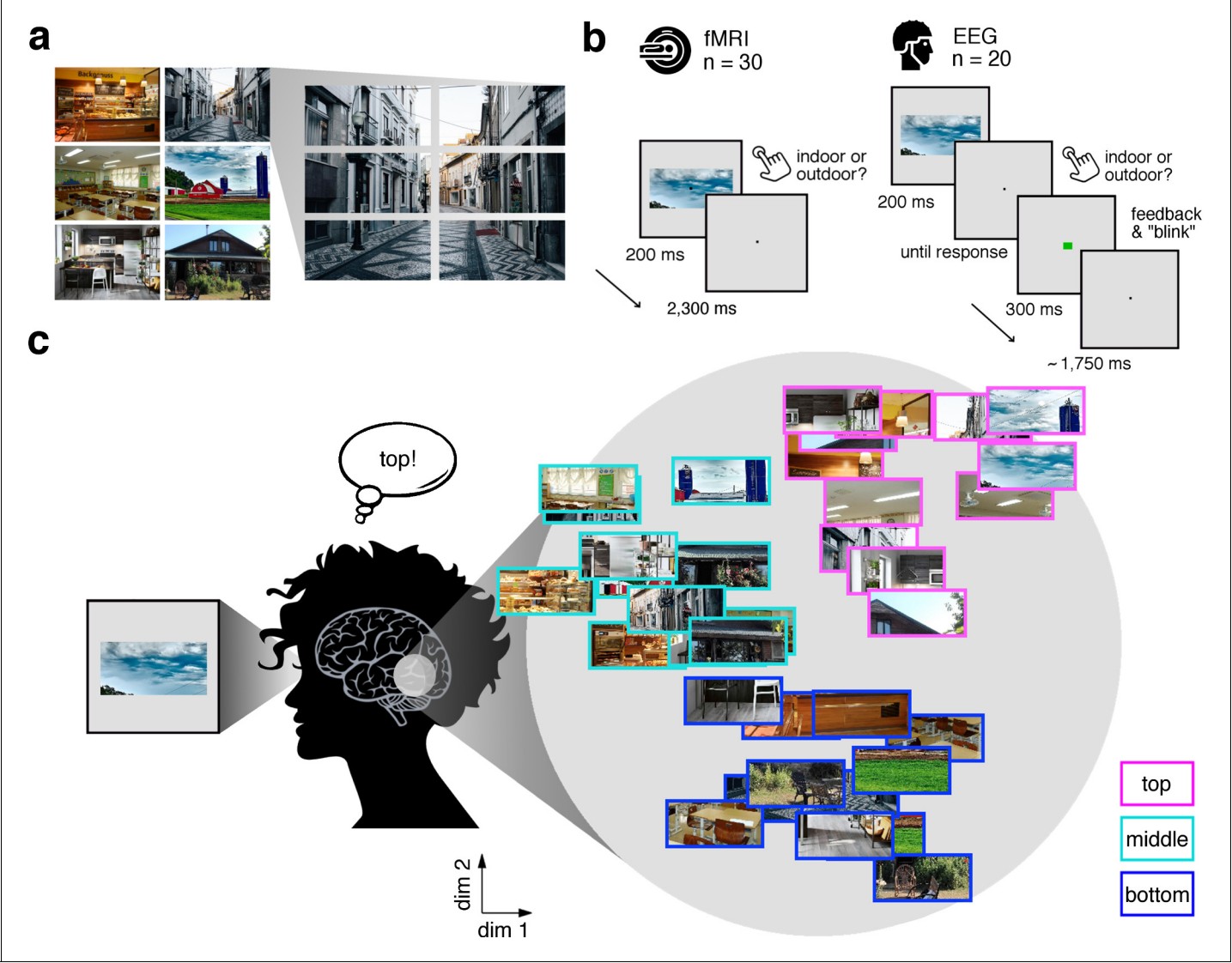

**Figure 1.** Experimental design and rationale of schema-based information sorting. (a) The stimulus set consisted of six natural scenes (three indoor, three outdoor). Each scene was split into six rectangular fragments. (b) During the fMRI and EEG recordings, participants performed an indoor/outdoor categorization task on individual fragments. Notably, all fragments were presented at central fixation, removing explicit location information. (c) We hypothesized that the visual system sorts sensory input by spatial schemata, resulting in a cortical organization that is explained by the fragments' within-scene location, predominantly in the vertical dimension: Fragments stemming from the same part of the scene should be represented similarly. Here we illustrate the hypothesized sorting in a two-dimensional space. A similar organization was observed in multi-dimensional scaling solutions for the fragments' neural similarities (see *Figure 1—figure supplement 1* and *Video 1*). In subsequent analyses, the spatiotemporal emergence of the schema-based cortical organization was precisely quantified using representational similarity analysis (*Figure 2*).

DOI: https://doi.org/10.7554/eLife.48182.002

The following figure supplement is available for figure 1:

**Figure supplement 1.** MDS visualization of neural RDMs 34 MDS visualization of neural RDMs.

DOI: https://doi.org/10.7554/eLife.48182.003

location and category information accounted for cortical responses in the three ROIs and across time.

The key observation was that the fragments' vertical location predicted neural representations in OPA (t[29] = 4.12, p<0.001, $p_{corr}$ <0.05), but not in V1 and PPA (test statistics for all analyses and ROIs are reported in *Supplementary file 1*) (*Figure 2d*) and between 55 ms and 685 ms (peak: t[19] = 9.03, p<0.001, $p_{corr}$ <0.05) (*Figure 2e*). This vertical-location organization was consistent across

t = −200 ms

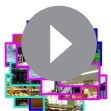

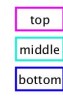

top
middle
bottom

**Video 1.** Time-resolved MDS visualization of the neural RDMs. To directly visualize the emergence of schematic coding from the neural data, we performed a multi-dimensional scaling (MDS) analysis, where the time-resolved neural RDMs (averaged across participants) were projected onto a two-dimensional space. The RDM time series was smoothed using a sliding averaging window (15 ms width). Computing MDS solutions across time yielded a movie (5 ms resolution), where fragments travel through an arbitrary space, eventually forming a meaningful organization. Notably, around 200 ms, a division into the three vertical locations can be observed.
DOI: https://doi.org/10.7554/eLife.48182.004

the first and second half of the experiments (see *Figure 2—figure supplement 6*) and across all pairwise comparisons along the vertical axis (see *Figure 2—figure supplement 7*). No effects were observed for horizontal location, consistent with more rigid spatial scene structure in the vertical dimension (*Mandler and Parker, 1976*). This result provides a first characterization of where and when incoming information is organized in accordance with scene schemata: in OPA and rapidly after stimulus onset, scene fragments are sorted according to their origin within the environment.

The schema-based organization co-exists with a prominent scene-category organization: In line with previous findings (*Lowe et al., 2018*; *Walther et al., 2009*), category was accurately predicted in OPA (t[29] = 3.12, p=0.002, $p_{corr}$ <0.05) and PPA (t[29] = 4.26, p<0.001, $p_{corr}$ <0.05) (*Figure 2d*), and from 60 ms to 775 ms (peak: t[19] = 6.39, p<0.001, $p_{corr}$ <0.05) (*Figure 2e*).

To efficiently support vision in dynamic natural environments, schematic coding needs to be flexible with respect to visual properties of specific scenes. The absence of vertical location effects in V1 indeed highlights that schematic coding is not tied to the analysis of simple visual features. To more thoroughly probe this flexibility, we additionally conducted three complementary analyses (*Figure 3*).

First, we tested whether schematic coding is tolerant to stimulus features relevant for visual categorization. Categorization-related features were quantified using a deep neural network (DNN; ResNet50), which extracts such features similarly to the brain (*Wen et al., 2018*). We removed DNN features by regressing out layer-specific RDMs constructed from DNN activations (see Materials and Methods for details) (*Figure 3a*); subsequently, we re-estimated location and category information.

After removing DNN features, category information was rendered non-significant in both fMRI and EEG signals. When directly comparing category information before and after removing the DNN features, we found reduced category information in PPA (t[29] = 2.48, p = 0.010, $p_{corr}$ <0.05) and OPA (t[29] = 1.86, p = 0.036, $p_{corr}$ >0.05), and a strong reduction of category information across time, from 75 ms to 775 ms (peak t[19] = 13.0, p<0.001, $p_{corr}$ <0.05). Together, this demonstrates that categorization-related brain activations are successfully explained by DNN features (*Cichy et al., 2016*; *Cichy et al., 2017*; *Groen et al., 2018*; *Güçlü and van Gerven, 2015*; *Wen et al., 2018*), indicating the appropriateness of our DNN for modelling visual brain activations. Despite the suitability of our DNN model for modelling categorical brain responses, vertical location still accounted for the neural organization in OPA (t[29] = 2.37, p = 0.012, $p_{corr}$ <0.05) (*Figure 3b*) and between 75 ms and 335 ms (peak: t[19] = 5.06, p<0.001, $p_{corr}$ <0.05) (*Figure 3c*). Similar results were obtained using a shallower feed-forward DNN (see *Figure 3—figure supplement 1*). This result suggests that schematic coding cannot be explained by categorization-related features extracted by DNN models.

DNN features are a useful control for flexibility regarding visual features, because they cover both low-level and high-level features, explaining variance across fMRI regions and across EEG processing time (see *Figure 3—figure supplement 2*; see also *Cichy et al., 2016*; *Güçlü and van Gerven, 2015*). However, to more specifically control for low-level features, we used two commonly employed low-level control models: pixel dissimilarity and GIST descriptors (*Oliva and Torralba, 2001*). These models neither explained the vertical location organization nor the category organization in the neural data (see *Figure 3—figure supplement 3*). Finally, as an even stronger control of

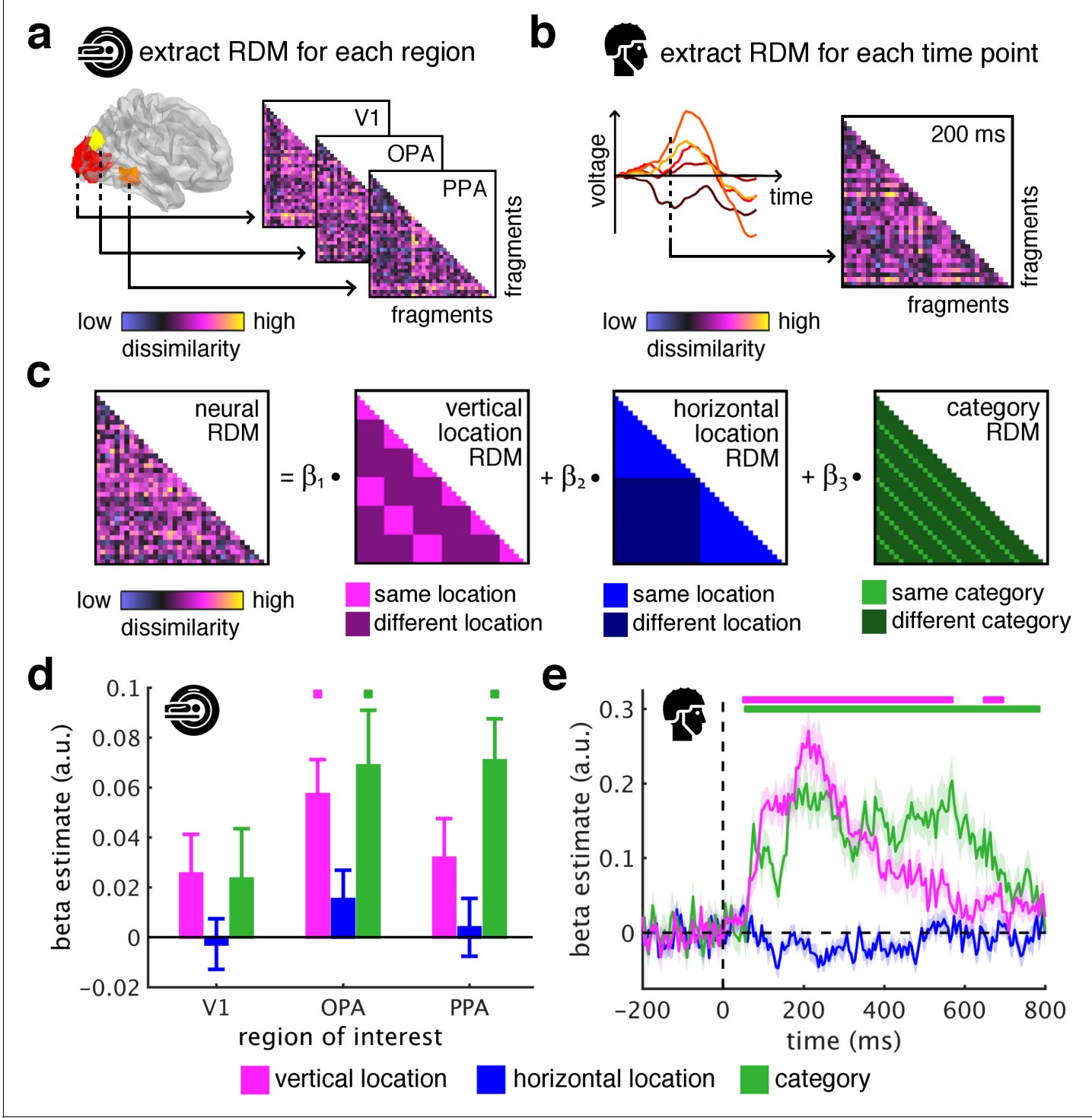

**Figure 2.** Spatial schemata determine cortical representations of fragmented scenes. (**a**) To test where and when the visual system sorts incoming sensory information by spatial schemata, we first extracted spatially (fMRI) and temporally (EEG) resolved neural representational dissimilarity matrices (RDMs). In the fMRI, we extracted pairwise neural dissimilarities of the fragments from response patterns across voxels in the occipital place area (OPA), parahippocampal place area (PPA), and early visual cortex (V1). (**b**) In the EEG, we extracted pairwise dissimilarities from response patterns across electrodes at every time point from −200 ms to 800 ms with respect to stimulus onset. (**c**) We modelled the neural RDMs with three predictor matrices, which reflected their vertical and horizontal positions within the full scene, and their category (i.e., their scene or origin). (**d**) The fMRI data revealed a vertical-location organization in OPA, but not V1 and PPA. Additionally, the fragment's category predicted responses in both scene-selective regions. (**e**) The EEG data showed that both vertical location and category predicted cortical responses rapidly, starting from around 100 ms. These results

*Figure 2 continued on next page*

*Figure 2 continued*

suggest that the fragments' vertical position within the scene schema determines rapidly emerging representations in scene-selective occipital cortex. Significance markers represent p<0.05 (corrected for multiple comparisons). Error margins reflect standard errors of the mean. In further analysis, we probed the flexibility of this schematic coding mechanism (*Figure 3*).

DOI: https://doi.org/10.7554/eLife.48182.005

The following figure supplements are available for figure 2:

**Figure supplement 1.** Details on neural dissimilarity construction.

DOI: https://doi.org/10.7554/eLife.48182.006

**Figure supplement 2.** fMRI response time courses.

DOI: https://doi.org/10.7554/eLife.48182.007

**Figure supplement 3.** Pairwise decoding across EEG electrode groups.

DOI: https://doi.org/10.7554/eLife.48182.008

**Figure supplement 4.** RSA using central electrodes.

DOI: https://doi.org/10.7554/eLife.48182.009

**Figure supplement 5.** RSA using anterior electrodes.

DOI: https://doi.org/10.7554/eLife.48182.010

**Figure supplement 6.** Vertical location effects across experiment halves.

DOI: https://doi.org/10.7554/eLife.48182.011

**Figure supplement 7.** Pairwise comparisons along the vertical axis.

DOI: https://doi.org/10.7554/eLife.48182.012

**Figure supplement 8.** Controlling for task difficulty.

DOI: https://doi.org/10.7554/eLife.48182.013

**Figure supplement 9.** Categorical versus Euclidean vertical location predictors.

DOI: https://doi.org/10.7554/eLife.48182.014

the low-level features encoded in V1, we used the neural dissimilarity structure in V1 (i.e., the neural RDMs) as a control model, establishing an empirical neural measure of low-level features. With V1 housing precise low-level feature representations, this measure should very well capture the features extracted during the early processing of simple visual features. However, removing the V1 dissimilarity structure did neither abolish the schematic coding effects in the OPA nor in the EEG data (see *Figure 3—figure supplement 3*). This shows that even if we had control models that approximated V1 representations extremely well – as well as the V1 representations approximate themselves – these models could not explain vertical location effects in downstream processing. Together, these results provide converging evidence that low-level feature processing cannot explain the schematic coding effects reported here.

Second, we asked whether schematic coding operates flexibly across visually diverse situations. To test this explicitly we restricted RDMs to comparisons between indoor and outdoor scenes, which vary substantially in visual characteristics (*Torralba and Oliva, 2003*) (*Figure 3d*).

Vertical location still predicted cortical organization in OPA (t[29] = 3.05, p = 0.002, $p_{corr}$ <0.05) (*Figure 3e*) and from 70 ms to 385 ms (peak: t[19] = 7.47, p<0.001, $p_{corr}$ <0.05) (*Figure 3f*). The generalization across indoor and outdoor scenes indicates that schematic coding operates similarly across radically different scenes, suggesting that the mechanism can similarly contextualize information across different real-life situations.

Finally, for a particularly strong test of flexibility, we tested for schematic coding after removing both DNN features and within-category comparisons (*Figure 3g*). In this analysis, OPA representations were still explained by the fragments' vertical location (t[29] = 2.38, p = 0.012, $p_{corr}$ <0.05) (*Figure 3h*). Notably, early schema effects were rendered non-significant, while vertical location still predicted representations after 180 ms (peak: t[19] = 4.41, p<0.001, $p_{corr}$ <0.05) (*Figure 3i*), suggesting a high degree of flexibility emerging at that time. Interestingly, across all analyses, vertical location information was exclusively found in OPA and always peaked shortly after 200 ms (see *Supplementary file 2*), suggesting that schematic coding occurs during early perceptual analysis of scenes.

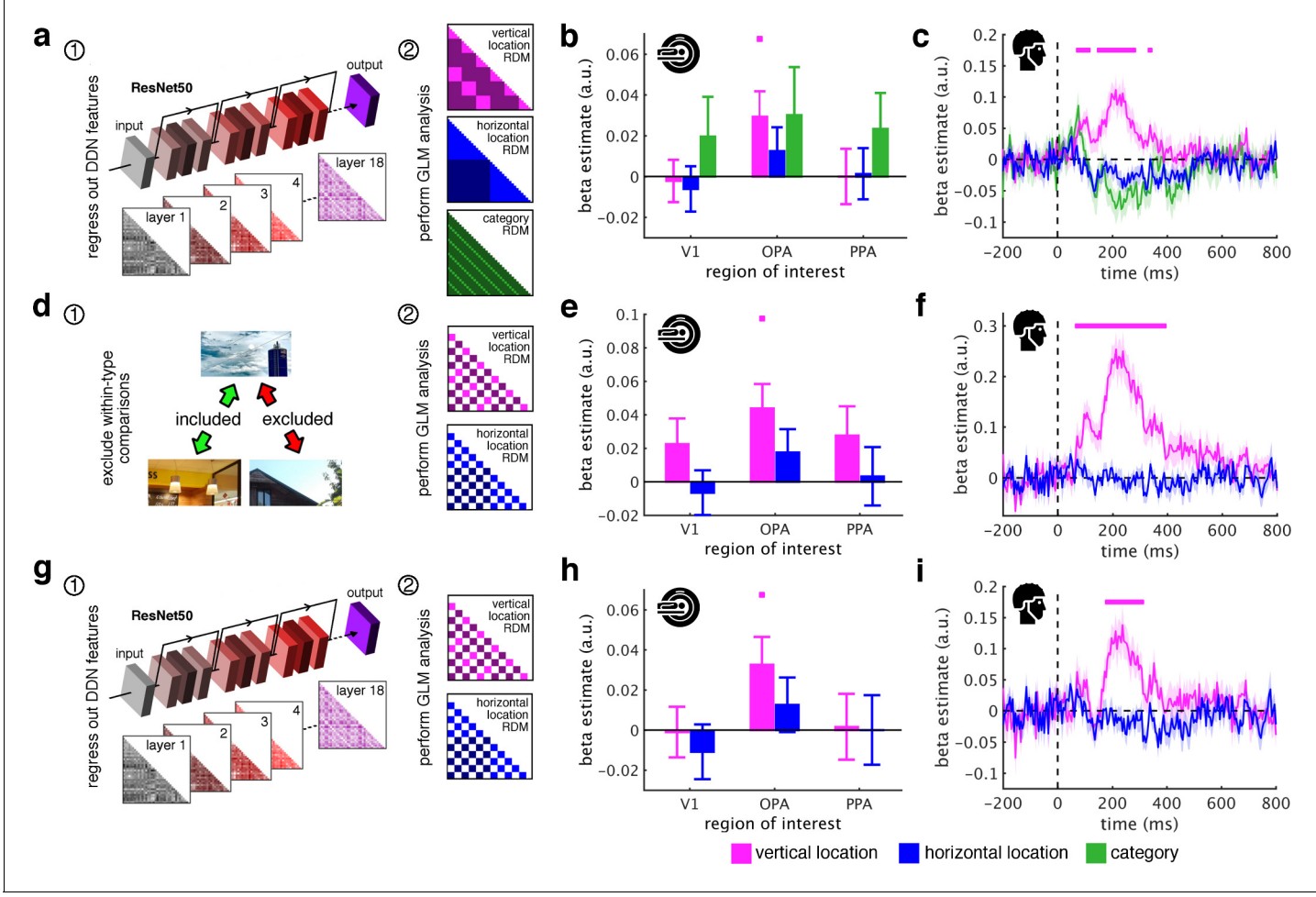

**Figure 3.** Schematic coding operates flexibly across visual and conceptual scene properties. (a) To determine the role of categorization-related visual features in this schematic organization, we regressed out RDMs obtained from 18 layers along the ResNet50 DNN before repeated the three-predictor general linear model (GLM) analysis (*Figure 2c*). (b/c) Removing DNN features abolished category information in fMRI and EEG signals, but not vertical location information. (d) To test for generalization across different scene types, we restricted location predictor RDMs to comparisons across indoor and outdoor scenes. Due to this restriction, category could not be modelled. (e/f) In this analysis, vertical location still predicted neural organization in OPA and from 70 ms. (g) Finally, we combined the two analyses: we first regressed out DNN features prior and then modelled the neural RDMs using the restricted predictor RDMs (d). (h) In this analysis, we still found significant vertical location information in OPA. (i) Notably, vertical location information in the EEG signals was delayed to after 180 ms, suggesting that at this stage schematic coding becomes flexible to visual and conceptual attributes. Significance markers represent p<0.05 (corrected for multiple comparisons). Error margins reflect standard errors of the mean.

DOI: https://doi.org/10.7554/eLife.48182.015

The following figure supplements are available for figure 3:

**Figure supplement 1.** AlexNet as a model of visual categorization.
DOI: https://doi.org/10.7554/eLife.48182.016
**Figure supplement 2.** DNN model fit.
DOI: https://doi.org/10.7554/eLife.48182.017
**Figure supplement 3.** Low-level control models.
DOI: https://doi.org/10.7554/eLife.48182.018

## Discussion

Together, our findings characterize a novel neural mechanism for contextualizing fragmented inputs during naturalistic vision. The mechanism exploits schemata to sort sensory inputs into meaningful representations of the environment. This sorting occurs during perceptual scene analysis in scene-selective OPA and within the first 200 ms of vision, and operates flexibly across changes in visual properties.

That schema-based coding can be localized to OPA is consistent with the region's important role in visual scene processing. Transcranial magnetic stimulation studies suggest that OPA activation is crucial for various scene perception tasks, such as scene discrimination (*Dilks et al., 2013*; *Ganaden et al., 2013*), navigating through scenes (*Julian et al., 2016*) and anticipating upcoming scene information (*Gandolfo and Downing, 2019*). Functional MRI work suggest that computations in the OPA include the analysis of spatial scene layout (*Dillon et al., 2018*; *Henriksson et al., 2019*; *Lowe et al., 2017*) and the parsing of local scene elements like objects and local surfaces (*Kamps et al., 2016*). Future studies are needed to clarify which of these computations mediate the schema-based coding described here.

As the current study is limited to a small set of scenes, more research is needed to explore whether schema-based coding generalizes to more diverse contents. It is conceivable that schema-based coding constitutes a more general coding strategy that may generalize to other visual contents (such as faces; *Henriksson et al., 2015*) and non-visual processing domains: when sensory information is fragmented and spatial information is unreliable, the brain may use schematic information to contextualize sensory inputs. This view is in line with Bayesian theories of perception where the importance of prior information for perceptual inference grows with the noisiness and ambiguity of the sensory information at hand (*Ernst and Banks, 2002*; *Kersten et al., 2004*).

The schema-based sorting of scene representations provides a mechanism for efficient communication between perceptual and cognitive systems: when scene information is formatted with respect to its role in the environment, it can be efficiently read out by downstream processes. This idea is consistent with the emerging view that cortical representations depend on functional interactions with the environment (*Bonner and Epstein, 2017*; *Groen et al., 2018*; *Malcolm et al., 2016*; *Peelen and Downing, 2017*). Under this view, formatting perceptual information according to real-world structure may allow cognitive and motor systems to efficiently read out visual information that is needed for different real-world tasks (e.g., immediate action versus future navigation). As the schema-based sorting of scene information happens already during early scene analysis, many high-level processes have access to this information.

Lastly, our results have implications for computational modelling of vision. While DNNs trained on categorization accurately capture the representational divide into different scene categories, they cannot explain the schema-based organization observed in the human visual system. Although this does not mean that visual features extracted by DNN models in principle are incapable of explaining schema-based brain representations, our results highlight that current DNN models of categorization do not use real-world structure in similar ways as the human brain. In the future, augmenting DNN training procedures with schematic information (*Katti et al., 2019*) may improve their performance on real-world tasks and narrow the gap between artificial and biological neural networks.

To conclude, our findings provide the first spatiotemporal characterization of a neural mechanism for contextualizing fragmented visual inputs. By rapidly organizing visual information according to its typical role in the world, this mechanism may contribute to the optimal use of perceptual information for guiding efficient real-world behaviors, even when sensory inputs are incomplete or dynamically changing.

## Materials and methods

**Key resources table**

| Reagent type (species) or resource | Designation | Source or reference | Identifiers | Additional information |
|---|---|---|---|---|
| Software, algorithm | CoSMoMVPA | *Oosterhof et al., 2016* | RRID:SCR_014519 | For data analysis |
| Software, algorithm | fieldtrip | *Oostenveld et al., 2011* | RRID:SCR_004849 | For EEG data preprocessing |
| Software, algorithm | MATLAB | Mathworks Inc. | RRID:SCR_001622 | For stimulus delivery and data analysis |
| Software, algorithm | Psychtoolbox 3 | *Brainard, 1997* | RRID:SCR_002881 | For stimulus delivery |

*Continued on next page*

*Continued*

| Reagent type (species) or resource | Designation | Source or reference | Identifiers | Additional information |
|---|---|---|---|---|
| Software, algorithm | SPM12 | www.fil.ion.ucl.ac.uk/ spm/software/spm12/ | RRID:SCR_007037 | For fMRI data preprocessing |

## Participants

Thirty adults (mean age 23.9 years, *SD* = 4.4; 26 females) completed the fMRI experiment and twenty (mean age 24.0 years, SD = 4.3; 15 females) completed the EEG experiment. All participants had normal or corrected-to-normal vision. They all provided informed consent and received monetary reimbursement or course credits for their participation. All procedures were approved by the ethical committee of the Department of Education and Psychology at Freie Universität Berlin (reference 140/2017) and were in accordance with the Declaration of Helsinki.

## Stimuli

The stimulus set (*Figure 1a*) consisted of fragments taken from three images of indoor scenes (bakery, classroom, kitchen) and three images of outdoor scenes (alley, house, farm). Each image was split horizontally into two halves, and each of the halves was further split vertically in three parts, so that for each scene six fragments were obtained. Participants were not shown the full scene images prior to the experiment.

## Experimental design

The fMRI and EEG designs were identical, unless otherwise noted. Stimulus presentation was controlled using the Psychtoolbox (*Brainard, 1997*; RRID:SCR_002881). In each trial, one of the 36 fragments was presented at central fixation (7° horizontal visual angle) for 200 ms (*Figure 1b*). Participants were instructed to maintain central fixation and categorize each stimulus as an indoor or outdoor scene image by pressing one of two buttons.

In the fMRI experiment, the inter-trial interval was kept constant at 2,300 ms, irrespective of the participant's response time. In the EEG experiment, after each response a green or red fixation dot was presented for 300 ms to indicate response correctness; participants were instructed to only blink after the feedback had occurred. Trials were separated by a fixation interval randomly varying between 1500 ms and 2000 ms.

In the fMRI, participants performed six identical runs. Within each run, each of the 36 scene fragments was shown four times, resulting in 144 trials. Additionally, each run contained 29 fixation trials, where only the central fixation dot was shown. Runs started and ended with brief fixation periods; the total run duration was 7:30 min. In the EEG, each of the 36 fragments was presented 40 times during the experiment, for a total of 1440 trials, divided into 10 runs. Three participants performed a shorter version of the experiment, with only 20 repetitions of each image (720 trials in total).

In both experiments, participants performed very well in the indoor/outdoor categorization task (fMRI: 94% correct, 658 ms mean response time, EEG: 96%, 606 ms). Differences in task difficulty across fragments were not related to the neural effects of interest (*Figure 2—figure supplement 8*).

## fMRI recording and preprocessing

MRI data was acquired using a 3T Siemens Tim Trio Scanner equipped with a 12-channel head coil. T2*-weighted gradient-echo echo-planar images were collected as functional volumes (TR = 2 s, TE = 30 ms, 70° flip angle, 3 mm$^3$ voxel size, 37 slices, 20% gap, 192 mm FOV, 64 × 64 matrix size, interleaved acquisition). Additionally, a T1-weighted image (MPRAGE; 1 mm$^3$ voxel size) was obtained as a high-resolution anatomical reference. During preprocessing, the functional volumes were realigned and coregistered to the T1 image, using MATLAB (RRID:SCR_014519) and SPM12 (www.fil.ion.ucl.ac.uk/spm/; RRID:SCR_014519).

## fMRI region of interest definition

We restricted our analyses to three regions of interest (ROIs). We defined scene-selective occipital place area (OPA; *Dilks et al., 2013*) and parahippocampal place area (PPA; *Epstein and Kanwisher, 1998*) using a functional group atlas (*Julian et al., 2012*). As a control region, we defined early visual

cortex (V1) using a probabilistic atlas (*Wang et al., 2015*). All ROIs were defined in standard space and then inverse-normalized into individual-participant space. For each ROI, we concatenated the left- and right-hemispheric masks and performed analyses on the joint ROI.

## EEG recording and preprocessing

The EEG was recorded using an EASYCAP 64-channel system and a Brainvision actiCHamp amplifier. The electrodes were arranged in accordance with the standard 10–10 system. The data was recorded at a sampling rate of 1000 Hz and filtered online between 0.03 Hz and 100 Hz. All electrodes were referenced online to the Fz electrode. Offline preprocessing was performed in MATLAB, using the FieldTrip toolbox (*Oostenveld et al., 2011*; RRID:SCR_004849). The continuous EEG data were epoched into trials ranging from 200 ms before stimulus onset to 800 ms after stimulus onset, and baseline corrected by subtracting the mean of the pre-stimulus interval for each trial and channel separately. Trials containing movement-related artefacts were automatically identified and removed using the default automatic rejection procedure implemented in Fieldtrip. Channels containing excessive noise were removed based on visual inspection. Blinks and eye movement artifacts were identified and removed using independent components analysis and visual inspection of the resulting components. The epoched data were down-sampled to 200 Hz.

## Representational similarity analysis

To model the representational structure of the neural activity related to our stimulus set, we used representational similarity analysis (RSA; *Kriegeskorte et al., 2008*). We first extracted neural RDMs separately for the fMRI and EEG experiments, and then used the same analyses to model their organization. To retrieve the fragments' position within the original scene, as well their scene category, we used a regression approach, where we modeled neural dissimilarity as a linear combination of multiple predictors (*Proklova et al., 2016*; *Proklova et al., 2019*).

### Constructing neural dissimilarity – fMRI

For the fMRI data, we used cross-validated correlations as a measure of pairwise neural dissimilarity. First, patterns for each ROI were extracted from the functional images corresponding to the trials of interest. After shifting the activation time course by 3 TRs (i.e., 6 s, accounting for the hemodynamic delay), we extracted voxel-wise activation values for each trial, from the TR that was closest to the stimulus onset on this trial (for results across 6 TRs with respect to trial onset, see *Figure 2—figure supplement 2*). To account for activation differences between runs, the mean activation across conditions was subtracted from each voxel's values, separately for each run. For each ROI, response patterns across voxels were used to perform multivariate analyses using the CoSMoMVPA toolbox (*Oosterhof et al., 2016*; RRID:SCR_014519). For each TR separately, we performed correlation-based (*Haxby et al., 2001*) multi-voxel pattern analyses (MVPA) for each pair of fragments. These analyses were cross-validated by repeatedly splitting the data into two equally-sized sets (i.e., half of the runs per set). For this analysis, we correlated the patterns across the two sets, both within-condition (i.e., the patterns stemming from the two same fragments and from different sets) and between-conditions (i.e., the patterns stemming from the two different fragments and from different sets). These correlations were Fisher-transformed. Then, we subtracted the within- and between-correlations to obtain a cross-validated correlation measure, where above-zero values reflect successful discrimination. This procedure was repeated for all possible splits of the six runs. Performing this MVPA for all pairs of fragments yielded a 36 × 36 representational dissimilarity matrix (RDM) for each ROI. RDMs' entries reflected the neural dissimilarity between pairs of fragments (the diagonal remained empty).

### Constructing neural dissimilarity – EEG

For the EEG data, we used cross-validated classification accuracies as a measure of pairwise neural dissimilarity. We thus constructed RDMs across time by performing time-resolved multivariate decoding analyses (*Contini et al., 2017*). RDMs were built by computing pair-wise decoding accuracy for all possible combinations of the 36 stimuli, using the CoSMoMVPA toolbox (*Oosterhof et al., 2016*). As we expected the highest classification in sensors over visual cortex (*Battistoni et al., 2018*; *Kaiser et al., 2016*), only 17 occipital and posterior sensors (O1, O2, Oz,

PO3, PO4, PO7, PO8, POz, P1, P2, P3, P4, P5, P6, P7, P8, Pz) were used in this analysis. We report results for other electrode groups in *Figure 2—figure supplements 3–5*. For each participant, classification was performed separately for each time point across the epoch (i.e., with 5 ms resolution). The analysis was performed in a pair-wise fashion: Linear discriminant analysis classifiers were always trained and tested on data from two conditions (e.g., the middle left part of the alley versus the top right part of the farm), using a leave-one-trial-out partitioning scheme. The training set consisted of all but one trials for each of the two conditions, while one trial for each of the two conditions was held back and used for classifier testing. This procedure was repeated until every trial was left out once. Classifier performance was averaged across these repetitions. The pairwise decoding analysis resulted in a 36-by-36 neural RDM for each time point. A schematic description of the RDM construction can be found in *Figure 2—figure supplement 1*.

## Location and category predictors

We predicted the neural RDMs in a general linear model (GLM; see below) with three different predictor RDMs (36 × 36 entries each) (*Figure 2c*): In the vertical location RDM, each pair of conditions is assigned either a value of 0, if the fragments stem from the same vertical location, or the value 1, if they stem from different vertical locations (for results with an alternative predictor RDM using Euclidean distances see *Figure 2—figure supplement 9*). In the horizontal location RDM, each pair of conditions is assigned either a value of 0, if the fragments stem from the same horizontal location, or a value of 1, if they stem from different horizontal locations. In the category RDM, each pair of conditions is assigned either a value of 0, if the fragments stem from the same scene, or a value of 1, if they stem from different scenes.

In an additional analysis, we sought to eliminate properties specific to either the indoor or outdoor scenes, respectively. We therefore constructed RDMs for horizontal and vertical location information which only contained comparisons between the indoor and outdoor scenes. These RDMs were constructed in the same way as explained above, but all comparisons within the same scene type of scene were removed (*Figure 3d*).

## Modelling neural dissimilarity

To reveal correspondences between the neural data and the predictor matrices, we used GLM analyses. Separately for each ROI (fMRI) or time point (EEG), we modelled the neural RDM as a linear function of the vertical location RDM, the horizontal location RDM, and the category RDM. Prior to each regression, the neural RDMs and predictor RDMs were vectorized by selecting all lower off-diagonal elements – the rest of the entries, including the diagonal, was discarded. Values for the neural RDMs were z-scored. Separately for each subject and each time point, three beta coefficients (i.e., regression weights) were estimated. By averaging across participants, we obtained time-resolved beta estimates for each predictor, showing how well each predictor explains the neural data over time.

Furthermore, we performed an additional GLM analysis with a vertical location predictor and a horizontal location predictor, where comparisons within indoor- and outdoor-scenes were removed (*Figure 3d–f*); these comparisons were also removed from the regression criterion. Using the same procedure as in the previous GLM analysis, we then estimated the beta coefficients for each predictor at each time point, separately for each subject. For this analysis, a category RDM could not be constructed, as all comparisons of fragments from the same scene were eliminated.

## Controlling for deep neural network features

To control for similarity in categorization-related visual features, we used a deep neural network (DNN) model. DNNs have recently become the state-of-the-art model of visual categorization, as they tightly mirror the neural organization of object and scene representations (*Cichy et al., 2016*; *Cichy et al., 2017*; *Cichy and Kaiser, 2019*; *Groen et al., 2018*; *Güçlü and van Gerven, 2015*; *Wen et al., 2018*). DNNs are similar to the brain as they are trained using excessive training material while dynamically adjusting the 'tuning' of their connections. Here, we used a DNN (see below) that has been trained to categorize objects across a large number of images and categories, therefore providing us with a high-quality model of how visual features are extracted for efficient categorization. By comparing DNNs activations and brain responses to the scene fragments, we could quantify

to which extent features routinely extracted for categorization purposes account for schema-based coding in the human visual system.

In a two-step approach, we re-performed our regression analysis after removing the representational organization emerging from the DNN. First, we used a regression model to remove the contribution of the dissimilarity structure in the DNN model. This model included one predictor for each layer extracted from the DNN (i.e., one RDM for each processing step along the DNN). Estimating this model allowed us to remove the neural organization explained by the DNN while retaining what remains unexplained (in the regression residuals). Second, we re-ran the previous regression analyses (see above), but now the residuals of the DNN regression were used as the regression criterion, so that only the organization that remained unexplained by the DNN was modeled.

As a DNN model, we used a pre-trained version (trained on image categorization for the ImageNet challenge) of the ResNet50 model (*He et al., 2016*), as implemented in MatConvNet (*Vedaldi and Lenc, 2015*). This model's deeper, residual architecture outperforms shallower models in approximating visual cortex organization (*Wen et al., 2018*). ResNet50 consists of 16 blocks of residual layer modules, where information both passes through an aggregate of layers within the block, and bypasses the block; then the residual between the processed and the bypassing information is computed. Additionally, ResNet50 has one convolutional input layer, and one fully-connected output layer. Here, to not inflate the number of intercorrelated predictor variables, we only used the final layer of each residual block, and thus 18 layers in total (16 from the residual blocks, and the input and output layers). For each layer, an RDM was built using 1-correlation between the activations of all nodes in the layer, separately for each pair of conditions. For regressing out the DNN RDMs, we added one predictor for each available RDM. In *Figure 3—figure supplement 1*, we show that an analysis using the AlexNet architecture (*Krizhevsky et al., 2012*) yields comparable results; in *Figure 3—figure supplement 2*, we additionally provide information about the DNN model fit across regions and time points.

## Statistical testing

For the fMRI data, we tested the regression coefficients against zero, using one-tailed, one-sample t-tests (i.e., testing the hypothesis that coefficients were greater than zero). Multiple-comparison correction was based on Bonferroni-corrections across ROIs. A complete report of all tests performed on the fMRI data can be found in *Supplementary file 1*. For the EEG data, we used a threshold-free cluster enhancement procedure (*Smith and Nichols, 2009*) to identify significant effects across time. Multiple-comparison correction was based on a sign-permutation test (with null distributions created from 10,000 bootstrapping iterations) as implemented in CoSMoMVPA (*Oosterhof et al., 2016*). The resulting statistical maps were thresholded at $Z > 1.64$ (i.e., $p<0.05$, one-tailed against zero). Additionally, we report the results of one-sided t-tests for all peaks effects. To estimate the reliability of onset and peak latencies we performed bootstrapping analyses, which are reported in Supplementary Items 2/3.

## Data availability

Data are publicly available on OSF (DOI.ORG/10.17605/OSF.IO/H3G6V).

## Acknowledgements

DK and RMC are supported by Deutsche Forschungsgemeinschaft (DFG) grants (KA4683/2-1, CI241/1-1, CI241/3-1). RMC is supported by a European Research Council Starting Grant (ERC-2018-StG).

## Additional information

### Funding

| Funder | Grant reference number | Author |
| --- | --- | --- |
| Deutsche Forschungsgemeinschaft | KA4683/2-1 | Daniel Kaiser |

| Deutsche Forschungsge-meinschaft | CI241/1-1 | Radoslaw M Cichy |
| --- | --- | --- |
| Deutsche Forschungsge-meinschaft | CI241/3-1 | Radoslaw M Cichy |
| H2020 European Research Council | ERC-2018-StG 803370 | Radoslaw M Cichy |

The funders had no role in study design, data collection and interpretation, or the decision to submit the work for publication.

### Author contributions

Daniel Kaiser, Conceptualization, Resources, Data curation, Software, Formal analysis, Supervision, Funding acquisition, Validation, Investigation, Visualization, Methodology, Writing—original draft, Project administration, Writing—review and editing; Jacopo Turini, Formal analysis, Investigation, Methodology, Writing—review and editing; Radoslaw M Cichy, Conceptualization, Supervision, Funding acquisition, Project administration, Writing—review and editing

### Author ORCIDs

Daniel Kaiser ⬤ https://orcid.org/0000-0002-9007-3160

### Ethics

Human subjects: All participants provided informed written consent. All procedures were approved by the ethical committee of the Department of Education and Psychology at Freie Universität Berlin (reference 140/2017) and were in accordance with the Declaration of Helsinki.

### Decision letter and Author response

Decision letter https://doi.org/10.7554/eLife.48182.026
Author response https://doi.org/10.7554/eLife.48182.027

## Additional files

### Supplementary files

• Supplementary file 1. Complete statistical report for fMRI results. The table shows test statistics and p-values for all tests performed in the fMRI experiment (*Figures 2* and *3*). Values reflect one-sided t-tests against zero. All p-values are uncorrected; in the main manuscript, only tests surviving Bonferroni-correction across the three ROIs (marked in color) are considered significant.
DOI: https://doi.org/10.7554/eLife.48182.019

• Supplementary file 2. Estimating peak latencies. The table shows means and standard deviations (in brackets) of peak latencies in ms for vertical location and category information in the main analyses (*Figures 2* and *3*). To estimate the reliability of peaks and onsets (*Supplementary file 3*) of location and category information in the key analyses, we conducted a bootstrapping analysis. For this analysis, we choose 100 samples of 20 randomly chosen datasets (with possible repetitions). For each random sample, we computed peak and onset latencies; we then averaged the peak and onset latencies across the 100 samples. Peak latencies were defined as the highest beta estimate in the time course. Notably, the peak latency of vertical location information remained highly stable across analyses.
DOI: https://doi.org/10.7554/eLife.48182.020

• Supplementary file 3. Estimating onset latencies. The table shows means and standard deviations (in brackets) of onset latencies in ms for vertical location and category information in the main analyses (*Figures 2* and *3*)). Onset latencies were quantified using the bootstrapping logic explained above (*Supplementary file 2*). Onsets were defined by first computing TFCE statistics for each random sample, with multiple-comparison correction based on 1000 null distributions. The onset latency for each sample was then defined as the first occurrence of three consecutive time points reaching significance (p<0.05, corrected for multiple comparisons).
DOI: https://doi.org/10.7554/eLife.48182.021

• Transparent reporting form DOI: https://doi.org/10.7554/eLife.48182.022

## Data availability

Data are publicly available on OSF (http://doi.org/10.17605/OSF.IO/H3G6V), as indicated in the Materials and Methods section of the manuscript.

The following dataset was generated:

| Author(s) | Year | Dataset title | Dataset URL | Database and Identifier |
|---|---|---|---|---|
| Kaiser D, Turini J, Cichy RM | 2019 | A neural mechanism for contextualizing fragmented information during naturalistic vision | http://doi.org/10.17605/OSF.IO/H3G6V | Open Science Framwork, 10.17605/OSF.IO/H3G6V |

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
