## [Decision Letter]

Thank you for submitting your article "A neural mechanism for contextualizing fragmented inputs during naturalistic vision" for consideration by *eLife*. Your article has been reviewed by three peer reviewers, one of whom is a member of our Board of Reviewing Editors, and the evaluation has been overseen by Joshua Gold as the Senior Editor. The reviewers have opted to remain anonymous.

The reviewers have discussed the reviews with one another and the Reviewing Editor has drafted this decision to help you prepare a revised submission.

This work combined fMRI, EEG and deep neural network model to investigate the neural basis of encoding of fragment location information (e.g., vertical, horizontal positions) across different scene categories. It addresses an important question about how the 'abstract' spatial layout that is not explicitly presented in the stimulus itself is represented in neural activities. The paper is clearly written and the approach and results are interesting and novel. However, there are some major issues brought up by the reviewers that need the authors to address and do additional analysis.

For a revision to be successful, you must address the following major issues:

Essential revisions:

1) The thirty-six figure segments come from only six natural images (3 indoor and 3 outdoor), which means the figure fragments would be repeatedly presented and be learned or memorized gradually. It is therefore hard to distinguish two interpretations – do the results reflect a true representation of spatial layout knowledge that would be automatically formed and could generalize to any natural images or do they derive from a learning and familiarization process after repeated exposure? A possible way to assess this is to divide the data into various stages and compare the early- and late-stage results. If the spatial layout knowledge is automatically represented regardless of learning, we would expect to see the same results in the early part. On the other hand, if it is indeed learning or memory process that induces the results, we would expect to see the pattern only in the late part but not in the early part.

2) Each scene picture was split into two halves horizontally and three parts vertically. Thus, there are confounding factors with regards to why the effect only occurred for vertical locations but not for horizontal locations. The authors should either collect new data or perform new analysis to address the issue.

3) The authors used DNN regression to confirm that the vertical position effect is not due to category-related information. However, the involvement of low-level features in discriminating vertical locations is still quite possible and could not be completely ruled out from the current analysis. For example, image segments at different vertical locations of natural scenes (upper, middle, lower) seem to be also associated with different low-level features (e.g., low spatial frequency for upper part, such as sky or ceiling, etc.). The authors could add additional analysis to clarify the confounds, for example, by creating a low-level dissimilarity design matrix, which would then predict involvement of V1 for the low-level features but not for vertical location, while the reverse for OPA.

4) It is hard to understand what exactly DNN features were removed. ResNet50 and AlexNet were used widely but these DNN models were trained by a very large set of images, whereas the present study only compares 6 specific images. The specific features to differentiate the 6 specific images may not be the same as the removed DNN features. The authors could show reconstructed images with DNN features removed and it is quite possible that human observes would still differentiate the 6 reconstructed images even when the DNN features are removed.

5) It is difficult to figure out what the authors were arguing was the mechanism or the consequence: Does knowledge/schema help sort incomplete information, or does the brain sort incomplete information so that we can extract knowledge? The paper is motivated by the former ("…the brain uses prior knowledge about where information typically appears in a scene to meaningfully sort incoming information") but then ends by stating the latter ("This mechanism empowers the visual brain to efficiently extract meaning from dynamic real-world environments, where it is confronted with sequences of incomplete visual snapshots"). Please add clarifications.

[Editors' note: further revisions were requested prior to acceptance, as described below.]

Thank you for resubmitting your work entitled "A neural mechanism for contextualizing fragmented inputs during naturalistic vision" for further consideration at *eLife*.

The manuscript has been improved but there was still one issue that was misunderstood or not clearly addressed.

In the original third comment, the reviewer pointed out that the DNN regression analysis cannot convincingly rule out the involvement of low-level properties in vertical position effect and suggested to directly test the DNN features in V1 ("The authors could add additional analysis to clarify the confounds, for example, by creating a low-level dissimilarity design matrix, which would then predict involvement of V1 for the low-level features but not for vertical location, while the reverse for OPA."). The authors have performed additional analysis by using 3 low-level control models, but again just examined whether the vertical position effect existed after regression using these models. The results indeed are not quite convincing and it seems that the regression even could not disrupt category representations as the previous DNN model showed. To explicitly address the concern, as the reviewer originally suggested, the authors should perform a new analysis to show that the DNN model does account for low-level property representation in V1, which would then support the claim that DNN regression could remove the low-level features.

---

## [Author Response]

Essential revisions:1) The thirty-six figure segments come from only six natural images (3 indoor and 3 outdoor), which means the figure fragments would be repeatedly presented and be learned or memorized gradually. It is therefore hard to distinguish two interpretations – do the results reflect a true representation of spatial layout knowledge that would be automatically formed and could generalize to any natural images or do they derive from a learning and familiarization process after repeated exposure? A possible way to assess this is to divide the data into various stages and compare the early- and late-stage results. If the spatial layout knowledge is automatically represented regardless of learning, we would expect to see the same results in the early part. On the other hand, if it is indeed learning or memory process that induces the results, we would expect to see the pattern only in the late part but not in the early part.

This is an important point. We deliberately chose few scene fragments for the experiment to compute reliable neural RDMs from multiple image presentations. We are aware of this choice posing a limitation to the current study and we now explicitly acknowledge this limitation in the Discussion section:

“As the current study is limited to a small set of scenes, more research is needed to explore whether schema-based coding generalizes to more diverse contents.”

Given the limited number of stimuli, it is in principle possible that the fragments’ location-specific representations only emerged after many stimulus repetitions across the experiment. To exclude this possibility, we performed the suggested analysis and separately analyzed data from the first and second halves of both experiments. For these halves (fMRI: first three versus last three runs, EEG: first versus second half of trials), we re-performed the main analysis for these halves. Critically, we found a very similar pattern of results with no statistical differences between the first and second half of each experiment, suggesting that the effect cannot be explained by excessive learning during the experiment. The results of this analysis are reported in Figure 2—figure supplement 6.

2) Each scene picture was split into two halves horizontally and three parts vertically. Thus, there are confounding factors with regards to why the effect only occurred for vertical locations but not for horizontal locations. The authors should either collect new data or perform new analysis to address the issue.

We now performed analyses where we analyzed all pairwise comparisons along the vertical axis (i.e., top versus bottom, top versus middle, and middle versus bottom), so that in each analysis there were only fragments from two different vertical locations. These analyses reveal a consistent vertical location organization, replicating the overall effect for each pairwise comparison, while suggesting that some comparisons (the ones including the top fragments) may contribute more to the effect. Results from these analyses are reported in Figure 2—figure supplement 7.

3) The authors used DNN regression to confirm that the vertical position effect is not due to category-related information. However, the involvement of low-level features in discriminating vertical locations is still quite possible and could not be completely ruled out from the current analysis. For example, image segments at different vertical locations of natural scenes (upper, middle, lower) seem to be also associated with different low-level features (e.g., low spatial frequency for upper part, such as sky or ceiling, etc.). The authors could add additional analysis to clarify the confounds, for example, by creating a low-level dissimilarity design matrix, which would then predict involvement of V1 for the low-level features but not for vertical location, while the reverse for OPA.

Thank you for this suggestion. Although regressing out deep neural networks should also control for low-level features to a substantial degree (previous research has shown that early DNN layers correspond well with early visual processing and activations in V1; see Cichy et al., 2016; Güclü and van Gerven, 2015), we now added new analyses where we used three additional models that explicitly control for low-level features: a pixel dissimilarity model, GIST descriptors (Oliva and Torralba, 2001), and V1 dissimilarity (as a neural approximation of low-level features). We re-performed our regression analyses after regressing out RDMs obtained from each of these low-level models. These analyses show that also the low-level models could not explain the fragments’ vertical location organization; neither could they explain their categorical organization. This indicates that schematic coding cannot be accounted for by low-level features. The new control analyses are summarized in Figure 3—figure supplement 2.

4) It is hard to understand what exactly DNN features were removed. ResNet50 and AlexNet were used widely but these DNN models were trained by a very large set of images, whereas the present study only compares 6 specific images. The specific features to differentiate the 6 specific images may not be the same as the removed DNN features. The authors could show reconstructed images with DNN features removed and it is quite possible that human observes would still differentiate the 6 reconstructed images even when the DNN features are removed.

This is an interesting point. First, we would like to point out that the reason we used categorization DNNs here was primarily to have a good measure of features that are routinely extracted for successful categorization, which includes low-level images properties as well as high-level category-defining features. It is true that the DNNs used here were trained to extract these features from large image databases, of whose properties the few images used in the study only capture a small fraction. However, it is worth noting that this qualitatively resembles processing in the visual system: The types of features the visual system routinely extracts are defined by excessive experience with large varieties of inputs, but at any single moment, we only have a very limited amount of input from which we extract a subset of these features. We now stress the reasons for using pre-trained DNNs in the Materials and methods section:

“DNNs are similar to the brain as they are trained using excessive training material while dynamically adjusting the “tuning” of their connections. […] By comparing DNNs activations and brain responses to the scene fragments, we could quantify to which extent features routinely extracted for categorization purposes account for schema-based coding in the human visual system.”

Second, we would certainly predict that after removing DNN features from the images, participants would still be able to visually discriminate between the images. We conceive of this as a feature rather than as a problem – ultimately the location information observed in brain responses must stem from visual features of the images. The key finding here is that the feature organization in DNN models, despite their similarity to visual cortex representations, cannot account for the schematic coding observed in the brain. We now bring up this point in the Discussion:

“While DNNs trained on categorization accurately capture the representational divide into different scene categories, they cannot explain the schema-based organization observed in the human visual system. Although this does not mean that visual features extracted by DNN models in principle are incapable of explaining schema-based brain representations, our results highlight that current DNN models of categorization do not use real-world structure in similar ways as the human brain.”

Finally, we agree that the current analysis does not provide information on the exact features accounted for by the DNN. Reconstruction images after controlling for DNN features could indeed provide a useful avenue for delineating the features that are uniquely extracted by the DNN and scene-selective cortex, respectively. However, we think that this is beyond the scope of the current paper, because reliably defining these features would ultimately require a larger, more diverse set of images and the acquisition of new experimental data.

5) It is difficult to figure out what the authors were arguing was the mechanism or the consequence: Does knowledge/schema help sort incomplete information, or does the brain sort incomplete information so that we can extract knowledge? The paper is motivated by the former ("…the brain uses prior knowledge about where information typically appears in a scene to meaningfully sort incoming information") but then ends by stating the latter ("This mechanism empowers the visual brain to efficiently extract meaning from dynamic real-world environments, where it is confronted with sequences of incomplete visual snapshots"). Please add clarifications.

The ending did indeed not reflect our interpretation very clearly. We re-worded the concluding paragraph to make it more consistent with the Introduction:

“To conclude, our findings provide the first spatiotemporal characterization of a neural mechanism for contextualizing fragmented visual inputs. By rapidly organizing visual information according to its typical role in the world, this mechanism may contribute to the optimal use of perceptual information for guiding efficient real-world behaviors, even when sensory inputs are incomplete or dynamically changing.”

[Editors' note: further revisions were requested prior to acceptance, as described below.]

The manuscript has been improved but there was still one issue that was misunderstood or not clearly addressed.In the original third comment, the reviewer pointed out that the DNN regression analysis cannot convincingly rule out the involvement of low-level properties in vertical position effect and suggested to directly test the DNN features in V1 ("The authors could add additional analysis to clarify the confounds, for example, by creating a low-level dissimilarity design matrix, which would then predict involvement of V1 for the low-level features but not for vertical location, while the reverse for OPA."). The authors have performed additional analysis by using 3 low-level control models, but again just examined whether the vertical position effect existed after regression using these models. The results indeed are not quite convincing and it seems that the regression even could not disrupt category representations as the previous DNN model showed. To explicitly address the concern, as the reviewer originally suggested, the authors should perform a new analysis to show that the DNN model does account for low-level property representation in V1, which would then support the claim that DNN regression could remove the low-level features.

Thanks for giving us the opportunity to address the remaining comment in more detail. If we understand correctly, the main worry is that the DNN models used in our study are not good models of low-level feature coding in cortex (i.e., they so not match well with V1-level representations), and thus regressing out DNN features is not a convincing way of ruling out low-level features as the explanation behind the main effect of interest, i.e. the vertical location effect.

We strongly agree that controlling for low-level features is an important issue in the current study. We see that in the previous revision we did not make sufficiently clear the comprehensive way in which we control for the effect of low-level features, DNN feature regression being only one of them. We therefore expose our perspective in detail below and made changes to the manuscript that make our argumentation more explicit.

Our pieces of evidence against low-level features behind the main effect of interest are the following. Most directly, we show that cortical area V1 does not exhibit vertical location effects. The absence of the effect of interest in the core region in cortex coding for low-level features is often taken as strong evidence against an involvement of low-level features. This thinking applies in our study as well. We make this point explicit in the current version of the manuscript:

“To efficiently support vision in dynamic natural environments, schematic coding needs to be flexible with respect to visual properties of specific scenes. The absence of vertical location effects in V1 indeed highlights that schematic coding is not tied to the analysis of simple visual features.”

Naturally, we agree that more comprehensive and corroborative evidence is desirable. Therefore, we go beyond this single piece of evidence with a series of control analyses.

First, we show that only taking visually very different indoor and outdoor scenes into account does not remove vertical location effect. Thus, it is unlikely that the vertical location effect simply reflects low-level features.

Second, we remove DNN features (or features extracted by low-level control models such as the GIST descriptor) and find that this does not abolish the vertical location effects. The reviewer questions in particular whether our DNN model accurately captures low-level feature representations. We believe that it is a fair model – we show that the DNN does explain variance in all regions examined, including V1 (Figure 3—figure supplement 2).

The reviewer further notes that the V1 data (category and vertical location effects) look somewhat comparable before and after regressing out the control models and takes this as suggesting that these are bad models of V1. However, given that we show that the DNN does explain variance in V1, this is also parsimoniously explained by there being no meaningful category or location organization in the V1 data to begin with.

That being said, we fully agree with the reviewer that although the DNN used might be a fair model of V1-level features, it is certainly not the best possible model. Thus, one could still assume that if DNNs approximated low-level feature representations in V1 more faithfully (with more variance explained), removing these DNNs' features might abolish the vertical location organization in higher-level cortex (i.e., the OPA), too.

To exclude this possibility (i.e., that it's simply a very good match to low-level feature representations in V1 that's needed to remove the OPA organization), we conducted a third control analysis. In this analysis, we used the best possible match to the neural V1 data as a low-level model – the V1 data itself! Naturally, the empirical neural V1 data explains the data in V1 (Figure 3—figure supplement 3). Crucially, after removing the neural organization in V1, the vertical location effect in OPA still persisted, showing that even if we had a model that predicted V1 exceptionally well (as well as V1 predicts itself), this model could not account for the vertical location effect.

We believe that this analysis already offers both properties that the reviewer asks for: a very good model of V1-level representation (arguably the best possible match to the neural organization in V1), and a strong vertical location effect in higher-level regions when the model is regressed out.

Together, we thus believe that the remaining open question boils down to how well DNNs can, in principle, model V1 organization. This is an interesting topic, which surely needs investigation – but from our point of view without immediate implications for the current paper. After all, we empirically show that even if the DNN perfectly captured the V1 organization, our conclusions would hold.

We laid out this argument in the revised manuscript:

“DNN features are a useful control for flexibility towards visual features, because they cover both low-level and high-level visual features, explaining variance across fMRI regions and across EEG processing time (see Figure 3—figure supplement 2; see also Cichy et al., 2016; Gücli and van Gerven, 2015). […] Together, these results provide converging evidence that low-level feature processing cannot explain the schematic coding effects reported here.”